# Old Habits Die Hard: Kinematic Carryover Between Low- and High-Impact Tasks in Active Females

**DOI:** 10.3390/sports13060160

**Published:** 2025-05-25

**Authors:** Vaishnavi Vivek Chiddarwar, Katherine F. Wilford, Troy L. Hooper, C. Roger James, Karthick Natesan, Aaron Likness, Gesine H. Seeber, Phillip S. Sizer

**Affiliations:** 1Center for Rehabilitation Research, Texas Tech University Health Sciences Center, Lubbock, TX 79415, USA; 2Physical Therapy Program, Murphy Deming College of Health Sciences, Mary Baldwin University, Fishersville, VA 22939, USA; 3Division of Orthopedics at Campus Pius-Hospital Oldenburg, School of Medicine and Health Sciences, Carl von Ossietzky Universität Oldenburg, 26121 Oldenburg, Germany; 4Department of Orthopedics, University Medical Center Groningen, University of Groningen, 9700 RB Groningen, The Netherlands

**Keywords:** ACL injury, biomechanics, female athlete, knee

## Abstract

Background: Knee injury risk screening protocols predominantly employ high-impact tasks (HIT), but there is a need for low-impact movement screening alternatives. This study aimed to investigate kinematic carryover between low-impact tasks (LIT) and HIT. Methods: This study employed a cross-sectional design. Eighteen healthy, active females with no history of injury within the last six months, aged between 18–35 years completed three trials of LIT (stand-to-sit, single-leg stand-to-sit) and HIT (drop vertical jump, single-leg drop vertical jump). Hip and knee three-dimensional kinematics were evaluated during LIT and HIT. Pearson correlation analyses were used to assess kinematic relationships between LIT and HIT. A post-hoc exploratory analysis examined the consistency of kinematic directionality across tasks. Results: In the frontal plane, the dominant hip, dominant knee, and non-dominant knee during LIT demonstrated a strong positive correlation and directional consistency with the corresponding values during HITs (*p* < 0.001). In the transverse plane, non-dominant hip, dominant knee, and non-dominant knee kinematics during LITs demonstrated directional consistency and a strong positive correlation with respective kinematics during HITs (*p* < 0.001). Conclusion: The similarities in hip and knee kinematic patterns suggest that motor responses may generalize across varying task intensities. Thus, LITs may be a useful tool in early knee injury risk identification.

## 1. Introduction

Knee injuries account for 40% of the total injuries sustained during athletic activities [1]. The relative incidence of knee injuries, such as anterior cruciate ligament (ACL) injury, among the general population has exhibited a persistent trend wherein rates observed in females remain consistently higher [2]. As such, the use of knee injury risk screening protocols has flourished. Established ACL injury risk screening protocols, such as the Landing Error Scoring System (LESS) and its abbreviated version LESS-RT, incorporate high-impact tasks (HIT) that include jump landings for early ACL injury risk identification [3]. Other ACL screening tests include Clinic-Based Algorithms for jump landing, Observational Screening of Dynamic Knee Valgus (OSDKV), 2D-Camera Method, Star Excursion Balance Test, and Tuck Jump Assessment and Functional Movement Screen (FMS) [4]. Additionally, a recent scoping review identified six broad categories for ACL injury and reinjury screening, namely balance and postural control, gait- and running-related tests, joint laxity, joint morphology and anthropometrics, jump tests, and strength tests [5]. Despite their popularity, current screening tests have limited predictive abilities as there is considerable heterogeneity in the test procedures and results. Furthermore, a recent systematic review concluded that the validity of the FMS in identifying ACL injury risk in active females is questionable [5,6,7].

While existing screening tests replicate knee injury mechanisms, they are prone to redundancies and are often over reliant on HITs [8]. This makes their utility, administration, and external validity unclear [5,6,9]. Furthermore, current screening tests identify risky movement mechanics in female athletes only during sport-specific tasks. However, these sports-specific HITs may produce pain or increase likelihood of re-injury. Previous research defines HIT and low-impact tasks (LIT) as dynamic tasks that involve contrasting vertical impact force magnitudes. Daily activities such as walking are considered to be LIT, whereas sports-specific jump landings serve as HIT [10]. While LITs are used in the post-ACL injury rehabilitation process [11], there is minimal use of these tasks in injury risk screening protocols.

Therefore, further research is warranted to create alternative screening tools, including a LIT that mimics the mechanism of injury (MOI) associated with ACL tears. Sports-specific HITs may produce pain or increase re-injury likelihood. Given the need for low-impact alternatives in knee injury risk assessment, the purpose of this study was to explore kinematic carryover between LIT and HIT. We hypothesized there would be meaningful correlations between hip and knee motion produced during LIT with the same corresponding values produced during HIT. Understanding these relationships may contribute to future research on low-impact knee injury risk screening methods.

## 2. Materials and Methods

### 2.1. Study Design and Ethical Considerations

This study incorporated a cross-sectional design. It was conducted at the Clinical Biomechanics Laboratory, Department of Rehabilitation Sciences, School of Health Professions (SHP), Texas Tech University Health Sciences Center (TTUHSC), Lubbock, TX, USA. All study procedures adhered to the Helsinki Declaration’s ethical principles [12]. Consequently, study-related procedures were approved by the TTUHSC’s Institutional Review Board (IRB) for the Protection of Human Subjects (protocol code: L23-048, date of approval: 1 May 2023). Before enrollment, eligible subjects were informed about potential risks and benefits before signing the approved written informed consent form.

### 2.2. Sample Size

Based on a moderate-to-large effect size of d = 0.7 [13], a desired power of 80%, a pre-set 0.05-alpha level [14], and a surmised 25% dropout rate, the responsible IRB approved the inclusion of 20 subjects (Figure 1).

### 2.3. Inclusion and Exclusion Criteria

All subjects were recruited from the local University’s surrounding community population, including students and staff. To confirm eligibility, subjects completed a screening questionnaire. Subjects qualified for study participation if they fulfilled the following pre-defined criteria: (1) age between 18 and 35 years; (2) female; (3) ability to stand for 60 s; (4) ability to freely sit down on, as well as rise from, a chair with no back or armrests; (5) ability to perform a drop vertical jump (DVJ) without assistance; and (6) recreationally active (self-reported). Subjects’ ages were limited to 35 years, and only females were included to capture the young adult population wherein knee injury incidence is increasing [15,16]. Participants were excluded for the following reasons: (1) existing low back pain (LBP) or lower extremity pain (LEP); (2) self-reported history of LBP or LEP requiring medical management within the past year; (3) self-reported history of spine, abdominal, or lower extremity (LE) surgery within the past year; (4) participation in a jump-landing program within the last six months; (5) self-reported pregnancy; (6) diagnosed neurological or joint disease affecting the trunk or LE; (7) cognitive disorder that impedes understanding of simple instructions; (8) Body Mass Index > 30; and (9) any skin allergy preventing the use of kinematic marker adhesives.

### 2.4. Instrumentation and Materials

Hip and knee kinematic analyses were performed using an eight-camera motion capture system (Vicon Motion Systems Ltd., Centennial, CO, USA) using the Nexus 2.12 software with a sampling frequency of 100 Hz. A set of 34 retro-reflective markers attached to predefined anatomical landmarks at the lumbopelvic region and LE, according to the Clinical Biomechanics Laboratory Gait Analysis Program (GaitAP) marker set [17], were used to quantify the three-dimensional (3D) hip and knee angles.

### 2.5. Data Collection

Recruited subjects wore athletic clothing during testing. Because footwear can alter selected LE joint kinematics when landing, subjects were tested in standardized smooth-surfaced athletic footwear suitable for marker placement provided by the lab [18,19,20]. Kinematic markers were attached bilaterally at the following sites: anterior superior iliac spine, posterior superior iliac spine, upper and lower posterior and anterior thigh and leg (plates), lateral and medial knee, tibial tuberosity, lateral and medial ankle, heel, and head of first, second, and fifth metatarsals. Anthropometric measurements included: foot length and width, malleoli height, thigh and calf girth, anterior and posterior superior iliac spine thicknesses, body height and weight, pelvis width, pelvis depth, and buttocks depth. Before data collection, a static trial was recorded with subjects positioned in a neutral standing posture for system calibration purposes.

Following kinematic marker applications, subjects performed a brief 5-min warm-up consisting of light endurance exercises. Once warmed up, subjects performed three successful trials of each task under investigation [21]. The LIT involved a self-paced bilateral and single-leg transition from standing to sitting on a surface that was 100% tibial height [22]. The HIT involved a bilateral and single-leg DVJ from a 23 cm high block. For the bilateral DVJ, subjects stepped off the block at a self-selected pace. The DVJ involved participants leading with the non-dominant foot, landing on the force plate with both feet, and immediately jumping vertically as high as possible before landing on both feet again [23]. Similarly, for the single-leg DVJ, subjects stepped off the block with their dominant or non-dominant leg (in randomized order), immediately jumped vertically as high as possible, and landed on the same leg. The 3D hip and knee kinematics were collected during the event when the pelvis center of mass was lowest during both HIT and LIT [24].

Subjects were given a standardized 10-s break between trials and 5 min between each type of task [25]. A trial was deemed successful when the following were met: (1) task completion without balance loss; (2) feet landed and remained on the force plates; and (3) markers remained in view of the motion capture cameras. The task order was randomized.

### 2.6. Data Reduction

Reflective markers were tracked, reconstructed, and labeled for each trial. Collected data were then exported to MatLab (Version 9.5, R2018b, Mathworks, Inc., Natick, MA, USA) for further processing using GaitAP. Marker trajectories were filtered using a 6 Hz low-pass, 2-phase, fourth-order Butterworth digital filter [26]. GaitAP applied a six-degrees-of-freedom link segment model to the segments using marker position data. The static standing calibration trial was used to define the segmental coordinate systems of the trunk, pelvis, thighs, legs, and feet, and the locations of joint axes were calculated. Joint kinematics were calculated using an XYZ rotation sequence in order of flexion/extension, abduction/adduction, and internal/external rotation. Pelvis segment kinematics were calculated as the orientation of the segment relative to the laboratory (global) coordinate system using a ZYX (rotation, obliquity, tilt) rotation sequence. Hip and knee kinematics were calculated as the orientation of the distal relative to the proximal segment using direct kinematics [27]. The pelvis COM was calculated as the midpoint between the average hip joint center (HJC) (average of right and left) and L5-S1 disc center by modeling the pelvis as an elliptical cylinder. Each HJC was determined using Harrington’s (2007) method [28], and the L5-S1 joint center was calculated from the pelvis coordinates provided by Reynolds (1982) [29] and categorized for small female, medium male, and large male pelvises and Nissan’s (1986) L5-S1 disc dimensions [17,30] (Figure 2).

### 2.7. Statistical Analysis

Descriptive analyses were performed for subject demographics. Means, standard deviations, and 95% confidence intervals (95% CI) for 3D hip and knee ROM were calculated. Data normality assessment included histograms, Q-Q plots, box plots, standardized skewness and kurtosis, and Shapiro–Wilk statistic measures.

Bivariate linear correlation analyses used the Pearson product-moment coefficient (r) to examine the relationships between 3D hip and knee ranges of motion during the LIT and HIT (α = 0.05). No alpha adjustments were incorporated due to the exploratory aim of the study.

## 3. Results

### 3.1. Participants

Twenty female participants were tested. However, due to instantaneous body position leading to hidden marker trajectory information throughout the LIT and HIT, two subjects’ data had to be excluded, leaving 18 subjects’ data for final analyses. Participants’ mean age was 23.9 (SD 1.5) years, and their average height and weight were 167 (SD 0.1) cm 63.8 (SD 7.2) kg, respectively. In accordance with the dominant leg information provided, 3 left and 15 right lower limbs were used as the test LE during single-leg LIT and HIT.

### 3.2. Kinematic Descriptive Data Results

Data normality was accepted when at least three of the following parameters were met: (1) Shapiro–Wilk *p* > 0.05, (2) W-statistic > 0.80, and (3) skewness and (4) kurtosis values each within ±2.0. All 18 subjects’ hip and knee kinematic angles data met the pre-defined normal distribution criteria (Table 1 and Table 2).

### 3.3. Correlation Results

We hypothesized there would be correlations between hip and knee motion produced during LIT with the same corresponding values produced during HIT. We further observed some notable patterns in the direction of the kinematics during LIT and HIT and have reported them below. During a majority of LITs, the dominant hip and dominant and non-dominant knees demonstrated a strong positive correlation and directional consistency (i.e., whether the kinematic direction during LIT corresponded with the kinematic direction during HIT) with the frontal plane motion during HITs (*p* < 0.001; Table 3). However, while demonstrating a strong positive correlation, the direction of the dominant knee frontal plane motion during the single-leg stand-to-sit task was inconsistent with the dominant knee frontal plane during HITs (*p* < 0.001; Table 3).

In the transverse plane, a majority of the non-dominant hip and dominant and non-dominant knee kinematics during LITs demonstrated directional consistency and a strong positive correlation with respective kinematics during HITs (*p* < 0.001; Table 4).

## 4. Discussion

This study aimed to establish the association between 3-D hip and knee kinematics during different LITs and HITs. Our findings support an association between hip and knee kinematics during LIT and the corresponding values during HIT. Strong positive correlations were found between hip and knee kinematics in the frontal and transverse planes during LIT (STS and single-leg STS) and the corresponding values during HIT (DVJ and single-leg DVJ). These findings suggest there are similarities in hip and knee kinematics during the two types of tasks, which may inform knee injury risk screening strategies for active females. Furthermore, this consistency may indicate that certain joint movement responses are stable across varying task intensities. Thus, screening for hip and knee mechanics during LITs may be a valuable, low-risk method for identifying individuals at risk for injuries during high-impact activities.

A previous study reported a high correlation between knee and hip angles in males and females during a step-down task from two heights with corresponding values during a single leg squat [25]. Moreover, another study observed that the unilateral step down produced greater motion in the frontal and transverse plane at the ankle and hip, whereas the DVJ task produced greater knee abduction [31]. This study suggested that both tasks continue to be used as screening tools since sensorimotor control appeared to be challenged in different manners [31]. Similarly, females demonstrated late knee extension initiation and significantly greater upper lumbar spine flexion during a sit-to-stand transfer versus males [32,33]. A more upright posture is commonly associated with increased vertical ground reaction forces. This has implications for knee injuries as ligament strain is related to maximal load and timing of ground reaction forces [34]. Low-impact tasks have been used as assessment tools for various patient populations. The sit-to-stand transition has been previously used to determine the relationship between dynamic knee valgus and patellofemoral pain syndrome in females [35]. This study demonstrated that using LITs associated with daily activities for determining knee injury risk could be effective for early detection of poor movement mechanics. Similarly, LITs such as sit-stand-sit and sit-to-stand transitions have been used to detect movement deficits after ACL reconstruction [36,37]. The use of LITs as alternative assessment tools may provide clinicians with a lower impact modality versus commonly used HIT skills. Furthermore, existing evidence supports combining the use of LITs and HITs to enhance the effectiveness of knee injury risk management strategies, including prophylactic programs [38]. In summary, clinicians can use these LIT tools when the patient’s recovery is too early for an HIT approach, or if restraints in time, space, or equipment limit assessment [35].

As previously suggested, no significant correlations were found in the sagittal plane, and one must examine why this may have occurred. A relevant study showed that, when examining all planes of movement in the LE, the STS transition demonstrated the least amount of anterior-posterior knee translation (sagittal plane) compared to other functional activities [39]. The sex-based differences in sagittal plane ergonomics have been examined during other daily tasks, such as lifting, with females demonstrating less sagittal plane motion at the lumbar spine and hip [40]. Similarly, females exhibit lower peak knee extension angles during stair descent, which is considered to be a daily life LIT [41]. Moreover, females have produced less trunk flexion and greater trunk and pelvis rotation during a single-leg squat [42]. These outcomes may illuminate why our study may not have been able to detect a correlation between LIT and HIT kinematic responses in the sagittal plane, as females tend to use more frontal and transverse plane degrees of freedom during LITs. Follow-up investigations may further elucidate mechanisms that create such outcomes.

The STS transition may be a useful low-impact alternative to assess movement patterns associated with increased knee injury risk such as non-contact ACL injury risk [20,43,44]. This could lead to earlier intervention and tailored corrective exercises, focusing on improving kinematic patterns across all movement intensity levels. In addition to sport-specific applications, this study’s findings can be employed for movement characteristics in other populations. For example, demonstrating a sliding-forward strategy during STS (more flexion of ankle and knee joints) is commonly observed in females [45]. This strategy may contribute to increased stress on the knee joint, thus a greater vulnerability for knee injury risk in females who use that strategy [45]. Relatedly, females with osteoarthritis demonstrate more pronounced knee valgus angles during STS [46]. Finally, older females require higher relative vastus lateralis neural activity during the eccentric STS movement [47]. This demonstrates that females use a greater proportion of maximal voluntary neural activation [47]. Motor control, neuroplasticity, and neural activity have lately received emphasis in the literature, particularly in the context of knee injury risk mitigation [48]. Therefore, these differences can further inform the sex-based knee injury risk discrepancy.

It is important to consider this study’s limitations when interpreting the results. First, we did not directly measure joint kinetics produced during LIT and HIT to comment on the movements’ relative impact on the joints involved. Instead, we selected the tasks based on the definitions provided in previous studies [10]. Second, the study population was a homogeneous, convenience sample only comprising healthy, young, and active females. We did not control for the subjects’ elevated activity levels. Future studies should include appropriate criteria to discriminate the relationship between kinematics and LIT/HIT depending on physical activity/sport participation experience levels. Finally, this study only established correlations between frontal and transverse plane hip and knee kinematics during LIT and HIT. Therefore, the presented results do not allow for any inference of causation. While two participants were excluded due to missing marker trajectory data, the final sample size of eighteen participants still provided sufficient power for the planned analyses. The missing data were randomly distributed and not associated with specific participant characteristics or movement tasks, minimizing the risk of systematic bias. However, we acknowledge that the exclusion may slightly limit the generalizability of the findings and reduce the representation of inter-individual variability.

Physical therapy focusing on motor learning shows that task specificity is key [49]. However, this study’s findings propose a carry-over effect, suggesting LITs may be a valuable, less demanding alternative to optimize LE injury risk screening where appropriate. Future research should explore the correlations between motor responses and LIT/HIT across different age groups in both sexes and different physical and mental maturation stages to gain deeper insight into whether these groups exhibit similar movement behavior as the current study’s sample. Furthermore, future research should explore specific, well-accepted biomechanical profiles associated with other common injuries, such as patellofemoral pain syndrome and iliotibial band syndrome. Future research could further expand the sample to include more participants and apply multivariate models to account for confounders such as limb dominance and physical activity levels.

## 5. Conclusions

The study’s findings deliver deeper insights into the association between hip and knee kinematics during LIT and HIT. We found a strong positive relationship between hip and knee kinematics during LIT with the same corresponding values during HIT in females. Based on movement patterns that have been previously identified as risk indicators for knee injury, LITs may be a useful tool in early risk identification. The similarities in hip and knee kinematic patterns suggest that motor responses related to movement efficiency and control may generalize across varying task intensities. This generalization could imply that improvements in movement mechanics in low-impact settings may carry over to high-impact performance. Conversely, movement mechanics during LITs may signal motor deficiencies that might manifest during HITs. Future research is warranted to further validate these findings.

## Figures and Tables

**Figure 1 sports-13-00160-f001:**
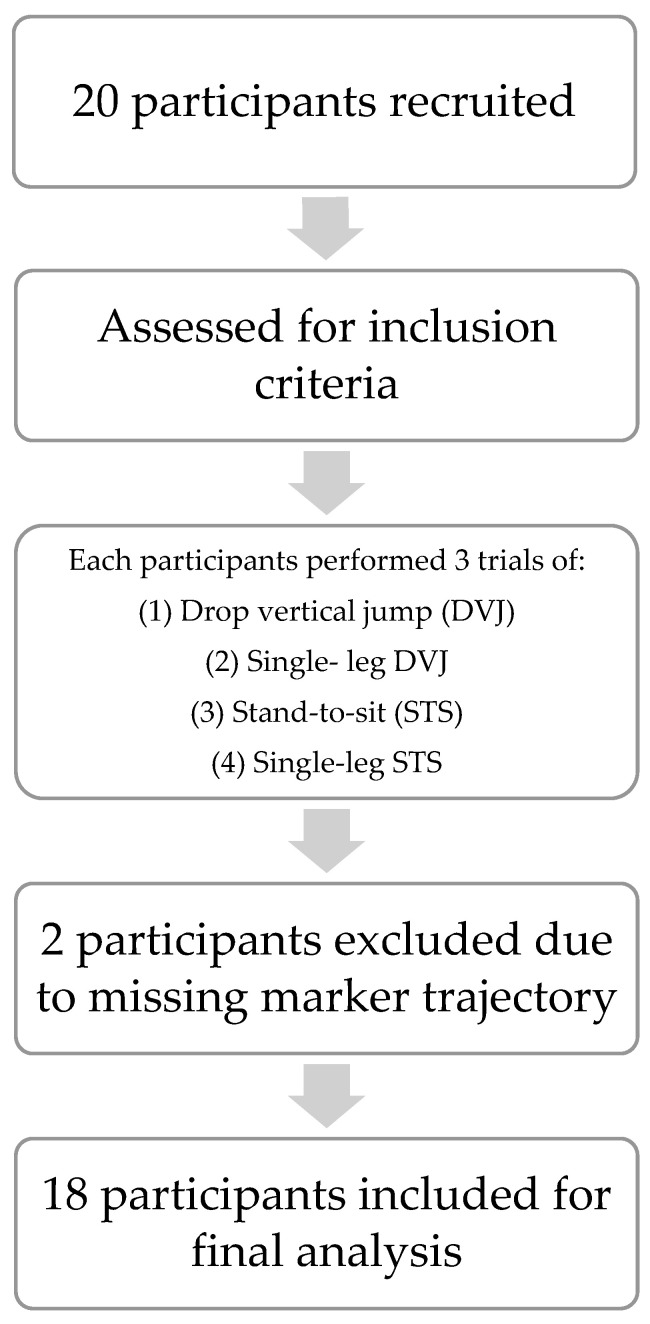
Flow diagram depicting research procedures.

**Figure 2 sports-13-00160-f002:**
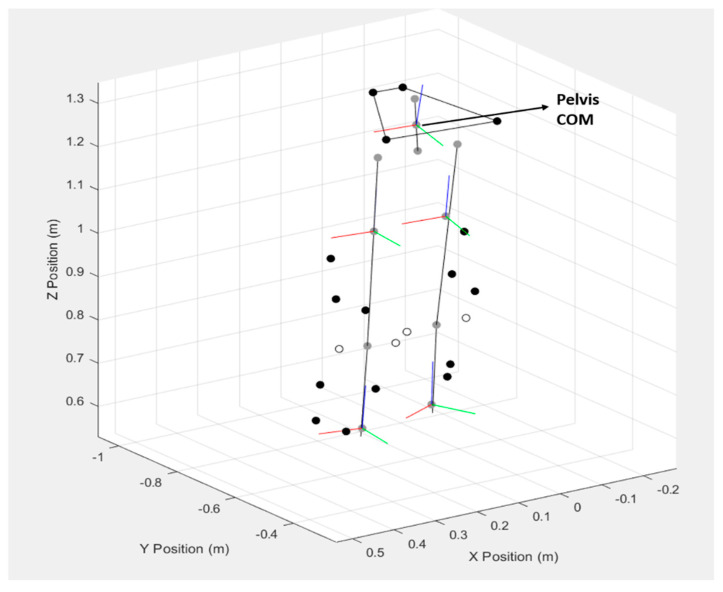
Stick figure depicting the pelvis center of mass.

**Table 1 sports-13-00160-t001:** Descriptive statistics for 3D hip and knee range of motion during bilateral LIT and HIT.

Activity	Joint	Plane	Mean	SD
bilateral DVJ	dominant hip	sagittal	99.77	14.86
		frontal	−5.27	7.35
		transverse	−5.84	6.80
	dominant knee	sagittal	99.99	12.57
		frontal	−6.89	7.08
		transverse	−12.19	8.73
	non-dominant hip	sagittal	102.27	14.55
		frontal	9.60	5.91
		transverse	5.57	7.55
	non-dominant knee	sagittal	101.38	14.34
		frontal	10.34	7.12
		transverse	6.45	8.93
bilateral STS	dominant hip	sagittal	69.07	14.63
		frontal	−2.75	6.00
		transverse	−2.93	6.94
	dominant knee	sagittal	85.89	8.03
		frontal	−3.22	8.72
		transverse	−15.03	7.66
	non-dominant hip	sagittal	69.75	13.56
		frontal	6.50	3.66
		transverse	3.68	6.54
	non-dominant knee	sagittal	84.62	9.78
		frontal	5.01	7.03
		transverse	6.67	7.26

Note: DVJ = Drop Vertical Jump; SD = Standard deviation; STS = stand-to-sit. Positive values indicate hip flexion, hip adduction, hip internal rotation, knee flexion, knee adduction, knee internal rotation.

**Table 2 sports-13-00160-t002:** Descriptive and normality statistics for 3D hip and knee range of motion during single-leg LIT and HIT.

Activity	Joint	Plane	Mean	SD
Single-leg DVJ	dominant hip	sagittal	68.89	13.31
		frontal	6.06	6.90
		transverse	−5.98	6.71
	dominant knee	sagittal	74.70	10.40
		frontal	−4.67	6.31
		transverse	−12.58	8.15
	non-dominant hip	sagittal	68.81	15.84
		frontal	−2.37	6.47
		transverse	9.78	6.32
	non-dominant knee	sagittal	72.8	10.06
		frontal	7.48	6.07
		transverse	2.84	9.31
Single-leg STS	dominant hip	sagittal	55.75	10.81
		frontal	−1.65	6.89
		transverse	−7.84	6.43
	dominant knee	sagittal	88.29	7.49
		frontal	1.09	6.80
		transverse	−13.01	8.10
	non-dominant hip	sagittal	57.62	11.98
		frontal	5.17	3.34
		transverse	10.04	5.45
	non-dominant knee	sagittal	88.59	9.87
		frontal	1.72	6.55
		transverse	6.06	8.20

Note: DVJ = drop vertical jump; SD = Standard deviation; STS = stand-to-sit. Positive values indicate hip flexion, hip adduction, hip internal rotation, knee flexion, knee adduction, and knee internal rotation.

**Table 3 sports-13-00160-t003:** Associations and directional consistency between frontal plane hip and knee kinematics during LIT with frontal plane hip and knee kinematics during HIT.

LIT	HIT	Joint	Kinematics LIT	Kinematics HIT	Relationship, *p*-Value	Consistency in the Kinematic Direction
STS	DVJ	Dominant Hip	Abduction	Abduction	r = 0.84, *p* < 0.01 **	Consistent
SL-STS	DVJ	Dominant Hip	Abduction	Abduction	r = 0.83, *p* < 0.01 **	Consistent
STS	DVJ	Dominant Knee	Abduction	Abduction	r = 0.81, *p* < 0.01 **	Consistent
STS	SL-DVJ	Non-dominant Knee	Adduction	Adduction	r = 0.71, *p* < 0.01 **	Consistent
SL-STS	DVJ	Dominant Knee	Adduction	Abduction	r = 0.80, *p* < 0.01 **	Not consistent
SL-STS	SL-DVJ	Dominant Knee	Adduction	Abduction	r = 0.86, *p* < 0.01 **	Not consistent
SL-STS	SL-DVJ	Non-dominant Knee	Adduction	Adduction	r = 0.85, *p* < 0.01 **	Consistent

Note: ** = *p* < 0.01; DVJ = Drop vertical jump; HIT = High-impact Tasks; LIT = Low-impact Tasks; SL = Single-leg; STS = Stand-to-Sit.

**Table 4 sports-13-00160-t004:** Associations and directional consistencies between transverse plane hip and knee kinematics during LIT with transverse plane hip and knee kinematics during HIT.

LIT	HIT	Joint	Kinematics LIT	Kinematics HIT	Relationship, *p*-Value	Consistency in the Kinematic Direction
STS	SL-DVJ	Non-dominant hip	Internal rotation	Internal rotation	r = 0.77, *p* < 0.01 **	Consistent
STS	SL-DVJ	Non-dominant knee	Internal rotation	Internal rotation	r = 0.76, *p* < 0.01 **	Consistent
STS	DVJ	Dominant knee	External rotation	External rotation	r = 0.77, *p* < 0.01 **	Consistent
SL-STS	SL-DVJ	Dominant knee	External rotation	External rotation	r = 0.82, *p* < 0.01 **	Consistent
SL-STS	SL-DVJ	Non-dominant knee	Internal rotation	Internal rotation	r = 0.69, *p* < 0.05 *	Consistent
SL-STS	DVJ	Non-dominant knee	Internal rotation	Internal rotation	r = 0.64, *p* < 0.05 *	Consistent

Note: * = *p* < 0.05; ** = *p* < 0.01; DVJ = Drop vertical jump; HIT = High-impact Tasks; LIT = Low-impact Tasks; SL = Single-leg; STS = Stand-to-Sit.

## Data Availability

Data are unavailable due to privacy and ethical restrictions. Yet, the raw data supporting this article’s conclusions will be made available by the authors upon reasonable request.

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
