# Peer review of "Old Habits Die Hard: Kinematic Carryover Between Low- and High-Impact Tasks in Active Females"

_sports, 2025, doi:10.3390/sports13060160_

Round 1

Reviewer 1 Report

Comments and Suggestions for Authors

This manuscript describes research comparing high and low impact tasks for testing knee stability in health female subjects.

The Introduction section provides a clear explanation of the tools currently used for assessing function following knee rehabilitation and the challenges to regaining full function and safe assessment of function.

The Methods section clearly explains the methodology used in this study. The assessments used are appropriate for the addressing the research question.

It is not clear why females only were included in the study. The authors state that lumbar spine flexion is different between males and females during sit-to-stand test. Why is this relevant when you only assessed females? (lines 227-228). The authors also make comparisons with studies assessing sex-based differences in sagittal plane ergonomics (lines 244-248).

The Results are also clearly presented.

The Discussion section is also well written and makes appropriate comparisons with previous research.

Author Response

Manuscript ID: sports-3524878

Dear Reviewer 1,

We are thankful for your time and effort in reading our manuscript. We appreciate your interest in our research and valuable feedback, which will strengthen this manuscript. Please see below our point-wise response to your comments, including the manuscript’s pages and line numbers where we have addressed them. All changes can be found in the marked manuscript in blue font, and have been included below for easier tracking. Please also find our responses to both reviewer's comments in the attached word file. 

REVIEWER #1 COMMENTS

Comment 1: This manuscript describes research comparing high and low impact tasks for testing knee stability in healthy female subjects.

Response: Thank you for reviewing our manuscript and for acknowledging the focus of our study. We appreciate the time and effort you have dedicated to evaluating our work. 

Comment 2: The Introduction section provides a clear explanation of the tools currently used for assessing function following knee rehabilitation and the challenges to regaining full function and safe assessment of function.

Response: Thank you. We appreciate your positive feedback on our introduction section.

Comment 3: The Methods section clearly explains the methodology used in this study. The assessments used are appropriate for the addressing the research question.

Response: Thank you.

Comment 4: It is not clear why females only were included in the study. The authors state that lumbar spine flexion is different between males and females during sit-to-stand test. Why is this relevant when you only assessed females? (lines 227-228). The authors also make comparisons with studies assessing sex-based differences in sagittal plane ergonomics (lines 244-248)

Response: Thank you for this comment. We especially chose young, active females due to their disproportionate vulnerability to knee injuries, such as non-contact ACL injuries. We have now specified that at the end of the inclusion criteria (Lines 93/94). We do agree that our discussion consists of comparative studies; however, we could not find relevant studies that only included females.

Comment 5: The Results are also clearly presented.The Discussion section is also well written and appropriately compares with previous research.

Response: We appreciate your comment on the quality of our results and discussion sections.

Reviewer 2 Report

Comments and Suggestions for Authors

1. Can you provide an experimental flowchart to make the experimental process, from inclusion of subjects to experimental protocols, clearer and more complete? Thank you.
2. In section 2.6, it is mentioned that the center of mass location used read the references, could you please provide a picture of this and indicate which sports biomechanics expert's model was used in the determination process? As the ethnicity may have varied in previous studies done. Thank you.
3. Can you explain in the discussion section how the missing data from the 2 subjects may have affected the results?
4. Can you provide a picture or schematic of the marker points pasted during the experiment? Thank you.
5. As a new way of predicting knee injuries, were the subjects themselves assessed for injury risk in this study, i.e., were they assessed for injury risk as an uninjured population by a certain gold standard?

Formatting issues
1. Please change the font of lines 26-29 and the font of the cited literature section. Thank you!
2. Please change the font in the tables, thank you.
3. In lines 189 192 195 for example (Results part), please mark with an asterisk of significance (* or **), as well as the tables. Thank you.
4. Please revise the numbers of references, some of them have the wrong font. Thank you.

Round 2

Reviewer 2 Report

Comments and Suggestions for Authors

1 Clearly justify your core hypothesis with existing biomechanical and motor control theories. Explain why LIT tasks might reflect HIT kinematics beyond correlation alone.

2 Reconsider the use of sit-to-stand and stand-to-sit tasks as LITs. Provide justification for their relevance to ACL injury mechanisms, or consider supplementing with more dynamic but low-impact tasks (e.g., single-leg squats, lunges).

3 Clearly state the generalizability limits due to a homogeneous sample. If possible, expand participant demographics (e.g., include males, different activity levels).
4 Randomize or counterbalance the order of LIT and HIT tasks in future studies. If already done, state this explicitly in the methods.
5 Move beyond simple correlation. Add multivariate models or regression to adjust for confounders (e.g., BMI, limb dominance). Consider applying Bonferroni or FDR corrections for multiple testing.
6 Soften your conclusion. Correlation does not imply causation or predictive utility. Avoid implying LITs can replace HITs in injury screening without longitudinal or diagnostic validation.
7 Provide a more comprehensive biomechanical rationale for the absence of sagittal plane correlations. Discuss potential task-specific strategies or gender-based movement patterns.
8 Define what is meant by “directional consistency” and evaluate it using established statistical techniques such as ICC or vector similarity indices.
9 Even without EMG data, discuss the role of neuromuscular control, preactivation, and proprioception in the observed kinematics. This adds depth and physiological context.
10 Propose how LIT kinematics might be used in real-world screening (e.g., with cutoff values, high-risk movement patterns, simple scoring systems).

Author Response

Manuscript ID: sports-3524878

Dear Editors and Reviewer 2,

We are thankful for the time and effort you put into reading our revised manuscript. We appreciate your interest in our research and valuable feedback, which will further strengthen this manuscript. Please see below our point-wise response to reviewer 2’s second round of comments including the manuscript’s line numbers where we have addressed them (these line numbers refer to the word file of the revised manuscript). All changes can be found in the marked manuscript in blue font for easier tracking

REVIEWER #2 ROUND #2 COMMENTS

Comment 1: Clearly justify your core hypothesis with existing biomechanical and motor control theories. Explain why LIT tasks might reflect HIT kinematics beyond correlation alone.

Author Response: Thank you for this suggestion. Regarding justification for our hypotheses, please refer to Lines 68-71 in the Introduction. Due to the inherent similarities between the LIT and HIT, specifically the fact that they both include the lowering of body weight in the upright position, we sought to explore the relationship between task performance.

Comment 2: Reconsider the use of sit-to-stand and stand-to-sit tasks as LITs. Provide justification for their relevance to ACL injury mechanisms, or consider supplementing with more dynamic but low-impact tasks (e.g., single-leg squats, lunges).

Author Response: Thank you for this feedback, however, since the study has already been completed, we cannot change the tasks performed.

Comment 3: Clearly state the generalizability limits due to a homogeneous sample. If possible, expand participant demographics (e.g., include males, different activity levels).

Author Response: As an exploratory manuscript, we agree that this study is not without several limitations, which are summarized in Lines 295-321. We are unable to further expand our participants for the current study as the study is closed. However, a future study with an expanded participant population is currently underway in our laboratory. We included this information in Lines 319-321.

Comment 4: Randomize or counterbalance the order of LIT and HIT tasks in future studies. If already done, state this explicitly in the methods.

Author Response: Thank you for this feedback, The order of task performance was randomized. The Methods section was modified to reflect this randomization. [Lines 137-138]

Comment 5: Move beyond simple correlation. Add multivariate models or regression to adjust for confounders (e.g., BMI, limb dominance). Consider applying Bonferroni or FDR corrections for multiple testing.

Author Response: Thank you for this feedback. We agree this would be an interesting exploration for future research, however, was outside the scope of the current project. As a result, the Discussion on future research has been edited. [Lines 319-321]

Comment 6: Soften your conclusion. Correlation does not imply causation or predictive utility. Avoid implying LITs can replace HITs in injury screening without longitudinal or diagnostic validation.

Author Response: We agree that correlation does not imply causation or predictive utility, currently stated in Lines 302-305. The scope of the current project was more exploratory in nature to test the hypothesis that there is a relationship between the performance of HIT and LIT. However, we have made edits to soften the conclusion as recommended in Lines 332-333.

Comment 7: Provide a more comprehensive biomechanical rationale for the absence of sagittal plane correlations. Discuss potential task-specific strategies or gender-based movement patterns.

Author Response: We found the lack of correlation in the sagittal plane to be an interesting finding from the current study. As such, we dedicated a significant part of the Discussion to this finding [Lines 265-278]. To summarize our rationale for this finding, we proposed that females tended to use more degrees of freedom in the frontal and transverse planes. If additional explanation is required, please provide additional feedback.    

Comment 8: Define what is meant by “directional consistency” and evaluate it using established statistical techniques such as ICC or vector similarity indices.

Author Response: Thank you for this question. Directional consistency was whether the kinematic direction was consistent between LIT and HIT. This term has now been clearly defined in Line 211-212. At this time, further statistical analysis of these data are not appropriate.

Comment 9: Even without EMG data, discuss the role of neuromuscular control, preactivation, and proprioception in the observed kinematics. This adds depth and physiological context.

Author Response: Thank you for this suggestion. We agree that the concepts listed by the reviewer would be interesting to explore further, however, they are beyond the scope of the current project.

Comment 10: Propose how LIT kinematics might be used in real-world screening (e.g., with cutoff values, high-risk movement patterns, simple scoring systems).

Author Response: Thank you for this suggestion, however, recommending this level of detail in the use of LIT in real-world screening is beyond the scope of the current project as additional research is required. The current study proposes that LIT performance may be useful for clinicians to detect high-risk movement patterns without the risk of HIT performance. Further details are contained Lines 279- 281.

Round 3

Reviewer 2 Report

Comments and Suggestions for Authors

Thank you for the author's reply and congratulations on the successful publication of the paper